# Generation of Bis(ferrocenyl)silylenes from Siliranes [note 1]

**DOI:** 10.3390/molecules25245917

**Published:** 2020-12-14

**Authors:** Yang Pan, Shogo Morisako, Shinobu Aoyagi, Takahiro Sasamori

**Affiliations:** 1Graduate School of Science, Nagoya City University, Nagoya, Aichi 467-8501, Japan; pan@nsc.nagoya-cu.ac.jp (Y.P.); aoyagi@nsc.nagoya-cu.ac.jp (S.A.); 2Division of Chemistry, Faculty of Pure and Applied Sciences, University of Tsukuba, Tsukuba, Ibaraki 305-8571, Japan; morisako.shogo.gf@u.tsukuba.ac.jp

**Keywords:** silirane, silylene, ferrocenyl group

## Abstract

Divalent silicon species, the so-called silylenes, represent attractive organosilicon building blocks. Isolable stable silylenes remain scarce, and in most hitherto reported examples, the silicon center is stabilized by electron-donating substituents (e.g., heteroatoms such as nitrogen), which results in electronic perturbation. In order to avoid such electronic perturbation, we have been interested in the chemistry of reactive silylenes with carbon-based substituents such as ferrocenyl groups. Due to the presence of a divalent silicon center and the redox-active transition metal iron, ferrocenylsilylenes can be expected to exhibit interesting redox behavior. Herein, we report the design and synthesis of a bis(ferrocenyl)silirane as a precursor for a bis(ferrocenyl)silylene, which could potentially be used as a building block for redox-active organosilicon compounds. It was found that the isolated bis(ferrocenyl)siliranes could be a bottleable precursor for the bis(ferrocenyl)silylene under mild conditions.

## 1. Introduction

Silylenes, i.e., divalent silicon analogues of carbenes, represent highly attractive reactive intermediates due to their considerable potential as building blocks for organosilicon compounds [1,2,3,4,5,6,7]. There are several examples of stable silylenes, which are usually stabilized thermodynamically by introduction of heteroatom-based substituents such as amino groups. For example, silicon analogues of *N*-heterocyclic carbenes (NHCs), *N*-heterocyclic silylenes (NHSis) [8,9,10,11,12,13,14,15,16,17], have been isolated as stable compounds (e.g., **I** [8,11] and **II** [9,13]) as well as the *N*-substituted cyclic silylene **III** [14] ([Fig molecules-25-05917-ch001]). However, such thermodynamically stabilized silylenes are too stable, i.e., they usually do not exhibit considerable reactivity as, e.g., building blocks for organosilicon compounds. Conversely, carbon-substituted silylenes, which do not contain a stabilizing electron-donating ligand, are highly reactive due to their considerable electrophilicity on account of their low-lying LUMOs [18,19,20,21,22,23,24]. Such highly reactive silylenes can be stabilized kinetically by sterically demanding substituents and subsequently being isolated under an inert atmosphere [25,26,27,28]. For example, dicoordinated, carbon-substituted silylenes [29,30,31,32] such as **IV** [29] and **V** [32] that bear sterically demanding substituents have been isolated and well characterized ([Fig molecules-25-05917-ch001]). However, it is very difficult to synthesize and isolate such silylenes as they usually require laborious and technically exigent synthetic procedures. In this context, easily accessible and bottleable precursors for carbon-substituted silylenes would be an attractive research target. In most cases, silylenes can be generated by reduction of a dihalosilane or photolysis of a trisilane [4,7,33]. When the substituents on the central silicon atom offer insufficient steric protection, the products from the aforementioned reactions usually afford complex mixtures of oligosilanes. Conversely, most cases of the reduction of sterically hindered dihalosilanes or the photolysis of trisilanes that bear bulky substituents result in the facile formation of a disilene, i.e., the formal dimer of a silylene [33,34,35,36,37,38,39]. To prepare a stable silylene precursor, it should thus be required to introduce very bulky substituents on the silicon atom to avoid facile dimerization of the generated silylenes [33]. On the other hand, too bulky substituents can be expected to negatively affect the inherent reactivity of the resulting silylene. In order to find the right balance in this trade-off relationship, transient silylenes with a combination of very bulky and less hindered groups have been designed. Our group has already reported the design and synthetic utility of the sterically demanding 2,5-bis(3,5-di-*t*-butylphenyl)-1-ferrocenyl (Fc*) group [40,41,42], and this group should be an appropriate substituent for bis(ferrocenyl)silylene **1** ([Fig molecules-25-05917-ch001]), which bears Fc* and ferrocenyl (Fc) groups on the silicon center [43]. Silylene **1** can be expected to (i) be very reactive due to its low-lying LUMO level, and (ii) show intriguing redox activity on account of its two ferrocenyl moieties.

## 2. Results and Discussion

Reaction of ferrocenyl (Fc) lithium with SiCl_4_ in toluene resulted in the formation of FcSiCl_3_ (**2**) [44], which was used in the following reactions without further purification due to its lability toward moisture. Subsequently, **2** was treated in Et_2_O at r.t. with Fc*Li (**3**) [40,41,42], which was prepared by the reaction of Fc*Br (Fc* = 2,5-bis(3,5-di(*t*-buthyl)phenyl)-1-ferrocenyl) with *n*-BuLi in Et_2_O, to give the corresponding bis(ferrocenyl)dichlorosilane (**4**) as a stable orange solid in 44% yield (Scheme 1). Although single crystals suitable for an X-ray diffraction (XRD) analysis could not be obtained due to partial hydrolysis that furnished Fc*FcSi(OH)_2_ [45], **4** was characterized by spectroscopic techniques.

As the reduction of diorganodichlorosilanes represents an efficient method for the generation of silylenes [4,33], we speculated that the reduction of bis(ferrocenyl)dichlorosilane **4** might furnish silylene **1**. However, the generation of **1** by the reduction of **4** was not observed when common metal reductants such as KC_8_, Na, or Li were employed, and only complex mixtures including Fc*H were obtained [41]. Conversely, when **4** was treated with lithium naphthalenide in THF at −78 °C, 2-naphthylsilane **5** was isolated in 30% yield, and its structure was unambiguously determined by spectroscopic and XRD analyses [46]. The formation of **5** should most likely be interpreted in terms of a C-H insertion of the reactive intermediate of silylene **1** with naphthalene. It should be noted here that it has previously been reported that isolable or transient silylenes react with aromatic compounds such as naphthalene and benzene to give the corresponding silepine or silirane derivatives via C-C insertion and [1+2]cycloaddition reactions, respectively [33,47,48,49,50]. Thus, the formation of **5** suggests the unique reaction manner of a silylene with an aromatic compound. Moreover, an isolable zwitterionic silylene has been reported to cleave a C-H bond of benzophenone via a 1,3-hydrogen shift [51,52].

At this point, we suspected that **5** was formed by the reaction of silylene **1** with naphthalene via the corresponding [1+2]cycloadduct, silirane **6**, which would be a conceivable intermediate in the reaction of a silylene with naphthalene. In order to clarify the reaction mechanism, the potential energy surfaces (PESs) of the reactions of silylene **1** with naphthalene were calculated at the B3PW91/6-311G(2d,p) level of theory (E_ZPE_: relative energies including zero-point energy corrections) [53]. The [1+2]cycloaddition of **1** with naphthalene was found to be slightly exothermic (ΔE_ZPE_ = 6.12 kcal/mol) to give silirane **6** with a barrier of ΔE_ZPE_
^‡^ = 21.6 kcal/mol. Unexpectedly, silirane **6** was not connected to **5** on the PES in these calculations. Instead, the highly exothermic (ΔE_ZPE_ = 37.4 kcal/mol) direct C-H insertion pathway of the reaction of **1** with naphthalene to furnish **5** was found to have a higher barrier of ΔE_ZPE_
^‡^ = 29.5 kcal/mol. The PES also showed that silirane **6** could give **1** with concomitant elimination of naphthalene via the reverse reaction of the formation of **6** (ΔE_ZPE_
^‡^ = 22.3 kcal/mol). Thus, it can be concluded **5** and **6** should be the thermodynamic and kinetic products, respectively, in the reaction of silylene **1** with naphthalene. At this stage, however, other plausible pathways for the formation of **5**, e.g., by the reaction of dichlorosilane **4** with lithium naphthalenide, which would not include the generation of silylene **1**, cannot be excluded from consideration (Scheme 2). However, based on the results of the theoretical calculations, silirane derivatives with a Si-C-C three-membered ring skeleton could potentially be considered appropriate precursors for silylenes such as **1**.

When a THF/Et_2_O solution of dichlorosilane **4** was reduced with sodium in the presence of an excess of cyclohexene and a small amount of naphthalene, a mixture of siliranes **9a** and **9b** was obtained (86% yield, **9a**:**9b** = 66:34). Even though it was difficult to purify each product by common separation techniques such as gel permeation chromatography (GPC), column chromatography, or recrystallization, a few crystals of **9a** and **9b** were obtained by recrystallization from hexane and pentane, respectively. Thus, siliranes **9a** and **9b** were fully characterized by spectroscopic analyses, while **9a** was structurally characterized by XRD analysis [46]. The similar up-field shifted ^29^Si NMR chemical shifts of **9a** (δ_Si_ = −67.2) and **9b** (δ_Si_ = −69.8) also indicated a silirane structure for **9b**, suggesting **9b** could be a stereoisomer of **9a** (Scheme 3). At this stage, **9a** and **9b** could not be separated from each other due the close similarity of their chemical properties. Heating a cyclohexene solution of a mixture of **9a** and **9b** (94:6 ratio) shifted the compositional ratio in favor of **9b** (**9a**:**9b** = 45:55), suggesting that **9a** and **9b** represent the kinetic and thermodynamic products in the reaction of **1** with cyclohexene, respectively. Both **9a** and **9b** are stable under ambient conditions in the solid state and in C_6_D_6_ solution. Interestingly, the conversion of **9a** to **9b** occurred in the solid state. After heating a mixture of solid **9a** and **9b** (ca. 74:26) at 120 °C for 30 min under reduced pressure, the ^1^H NMR spectrum of the solid dissolved in C_6_D_6_ showed a **9a**:**9b** ratio of ca. 45:55, suggesting a transformation of **9a** to **9b** even in the solid state.

The XRD structure of **9a** (Figure 1) shows that the cyclohexane moiety of **9a** is oriented toward the crowded space close to the Fc* group; **9b** could thus potentially exhibit a more stable geometry wherein the cyclohexyl group is oriented toward the less bulky ferrocenyl moiety. Theoretical calculations at B3PW91-D3(BJ)/6-311G(3d) level of theory suggest that **9b** is by 0.78 kcal/mol more stable than **9a**.

As described above, heating of the cyclohexene solution of siliranes **9a** and **9b** at 75 °C resulted in the conversion of **9a** to **9b** with concomitant formation of minor amounts of Fc*FcSiH(OH) (**10**) [46], suggesting the generation of silylene **1** in the equilibrium state. The formation of small amounts of **10** should most likely be interpreted in terms of a hydrolysis of silylene **1** generated by thermolysis of **9a** and/or **9b** due to the inevitable contamination with a small amount of moisture. Dissolving a mixture of **9a** and **9b** (66:34) in an excess amount of methanol at r.t. afforded **11** (28%), **12** (10%), and **13** (15%) as shown in Scheme 4 [6,46]. The formation of **11** suggests the generation of silylene **1** at r.t. However, siliranes **9a** and/or **9b** would probably undergo alcoholysis with MeOH to give **12** and **13** under these conditions. Thus, siliranes **9a** and **9b** could be appropriate thermal precursors for silylene **1** on heating, although they are sensitive toward protic solvents. Conversely, heating of a mixture of **9a** and **9b** (25:86) at 60 °C for 41 h in the presence of an excess of 2,3-dimethyl-1,3-butadiene afforded silolene **14** as the corresponding [1+4]cycloadduct of silylene **1** with 2,3-dimethyl-1,3-butadiene in 44% isolated yield, suggesting that the thermolysis of both **9a** and **9b** affords silylene **1** at this temperature. Thus, it can be concluded that siliranes **9a** and **9b** could work as synthetic precursors for bis(ferrocenyl)silylene **1** in such pericyclic reactions.

Subsequently, we performed theoretical calculations on the dissociation energies of siliranes bearing several organic substituents at B3PW91-D3(BJ)/6-311G(3d) level of theory (Scheme 5). The dissociation energies of **9a** and **9b** are ca. 17 kcal/mol, which are smaller than those of phenyl- and methyl-substituted siliranes **17** (26.0 kcal/mol) and **19** (27.6 kcal/mol). Considering the small dissociation energy of less bulky bis(ferrocenyl)silirane **15** (14.0 kcal/mol), the facile generation of silylene **1** from **9a** and **9b** under mild conditions could be explained, not by the steric hindrance due to the bulky Fc* group, but the electronic effect of the ferrocenyl groups. Even though the electronic perturbation from the ferrocenyl groups toward the silirane moiety of **9a** and **9b** remains unclear at present, the electron-donating properties of the ferrocenyl groups can be expected to lower the dissociation energy of the silirane skeleton.

## 3. Materials and Methods

### 3.1. General Information

All manipulations were carried out under an argon atmosphere using either Schlenk line techniques or glove boxes. All solvents were purified by standard methods. Trace amounts of water and oxygen remaining in the solvents were thoroughly removed by bulb-to-bulb distillation from potassium mirror prior to use. ^1^H, ^13^C, and ^29^Si-NMR spectra were measured on a JEOL ECZ-500R (^1^H: 500 MHz, ^13^C: 126 MHz, ^29^Si: 99 MHz) or on a Bruker AVANCE-400 spectrometer (^1^H: 400 MHz, ^13^C: 101 MHz, ^29^Si: 79 MHz). Signals arising from residual C_6_D_5_H (7.15 ppm) in C_6_D_6_ or CHCl_3_ (7.26 ppm) in CDCl_3_ were used as the internal standards for the ^1^H NMR spectra, while those of C_6_D_6_ (128.0 ppm) and CDCl_3_ (77.0 ppm) were used for the ^13^C NMR spectra, and external SiMe_4_ (0.0 ppm) for the ^29^Si NMR spectra. High-resolution mass spectra (HRMS) were obtained from a JEOL JMS-T100LP (DART) or JMS-T100CS (APCI) mass spectrometer (DART). Gel permeation chromatography (GPC) was performed on an LC-6AD (Shimadzu Corp., Kyoto, Japan) equipped with JAIGEL-1H and 2H (Japan Analytical Industry Co., Ltd., Tokyo, Japan) columns using toluene as the eluent. All melting points were determined on a Büchi Melting Point Apparatus M-565 and are uncorrected. Fc*Br (Fc* = 2,5-[3,5-(*t*Bu)_2_C_6_H_3_]_2_-1-ferrocenyl) was prepared according to literature procedures [40,41,42]. 1-Bromo-3,5-di-*tert*-butylbenzene was gifted by MANAC Inc., Tokyo, Japan.

### 3.2. Syntheses and Reactions

#### 3.2.1. Bis(ferrocenyl)dichlorosilane **4**

Ferrocene (1.00 g, 5.38 mmol) was dissolved in THF (4.0 mL) and cooled to 0 °C. During 30 min, a pentane solution of *t*-BuLi (3.6 mL, 1.6 M in pentane, 5.76 mmol) was added dropwise. Then, hexane (10 mL) was added to the reaction mixture, and the solution was kept at −78 °C for 15 min. The resulting orange precipitate including ferrocenyl lithium was filtered off, and then washed with small portions of hexane. The orange precipitate was then dissolved in toluene (4 mL), and the solution was added to SiCl_4_ (884 mg, 5.27 mmol). After stirring the mixture for 3 h at room temperature, the resulting inorganic salts were removed and the solvent of the filtrate was evaporated under reduced pressure to give ferrocenyltrichlorosilane **2** as an orange solid that was used for the subsequent reactions without further purification. An ether solution (2 mL) of Fc*Br (635 mg, 994 µmol) was treated with *n*-BuLi (0.5 mL, 2.60 M in hexane, 1.3 mmol) at 0 °C. After stirring for 6 h at room temperature, the solution of the resulting Fc*-Li (**3**) was added to **2** at 0 °C. After stirring for 12 h at room temperature, the solution was filtered and the filtrate was evaporated under reduced pressure. The obtained orange solid was subjected to GPC (toluene) to give bis(ferrocenyl)dichlorosilane **4** (366 mg, 434 µmol, 44%). **4**: orange solid. Mp. 107–118 °C. ^1^H-NMR (500 MHz, benzene-*d*_6_) δ 7.70 (d, *J* = 1.7 Hz, 4H), 7.49 (t, *J* = 1.7 Hz, 2H), 4.44 (s, 2H), 4.39 (s, 5H), 4.12 (s, 5H), 3.87 (s, 4H), 1.42 (s, 36H); ^13^C-NMR (126 MHz, benzene-*d*_6_) δ 150.0, 137.2, 128.3, 128.1, 127.9, 126.1, 121.3, 99.7, 75.1, 74.6, 72.0, 71.2, 69.8, 69.2, 64.5, 35.1, 31.8; ^29^Si-NMR (79 MHz, benzene-*d*_6_) δ 11.9. HRMS (APCI), *m/z*: Found: 844.23901 ([M^+^]), Calcd. for C_48_H_58_Cl_2_Fe_2_Si ([M^+^]): 844.23837.

#### 3.2.2. Bis(ferrocenyl)naphthylsilane **5**

A THF solution of LiNaph (0.2 mL, 1.1 M THF solution, 0.22 mmol) was added dropwise to a THF solution (0.2 mL) of bis(ferrocenyl)dichlorosilane **4** (47.7 mg, 56.5 µmol) at −78 °C. After stirring for 24 h at room temperature, the solvent was removed under reduced pressure, and the residue was dissolved in benzene. The resulting inorganic salts were removed by filtration and the filtrate was evaporated under reduced pressure to give **5** as an orange solid (quant. as judged by ^1^H-NMR spectrum). The orange solid of **5** was purified by GPC (toluene) to give **5** in pure form (15.1 mg, 16.7 µmol, 30%). **5**: orange solid. Mp. 195–200 °C. ^1^H-NMR (500 MHz, benzene-*d*_6_) δ 8.20 (s, 1H), 7.90 (d, *J* = 8.0 Hz, 1H), 7.74 (d, *J* = 1.7 Hz, 2H), 7.66 (d, *J* = 8.0 Hz, 1H), 7.63–7.64 (m, 1H), 7.61 (d, *J* = 1.7 Hz, 2H), 7.59 (q, *J* = 2.1 Hz, 1H), 7.42 (t, *J* = 1.7 Hz, 2H), 7.24–7.27 (m, 2H), 6.21 (s, 1H), 4.65 (d, *J* = 1.1 Hz, 2H), 4.34 (s, 5H), 4.16 (d, *J* = 1.1 Hz, 1H), 4.05 (s, 1H), 3.99 (s, 1H), 3.88 (s, 5H), 3.82 (d, *J* = 1.1 Hz, 1H), 1.37 (s, 18H), 1.20 (s, 18H);^29^Si-NMR (99 MHz, benzene-*d*_6_) δ—19.5. HRMS (APCI), *m/z*: Found: 902.36587 ([M^+^]), Calcd. for C_58_H_66_Fe_2_Si ([M^+^]): 902.36326.

#### 3.2.3. Siliranes **9a** and **9b**

A mixture of sodium (dispersion in mineral oil, 7.2 mg, 313 µmol), naphthalene (25.5 mg, 199 µmol), and cyclohexene (0.2 mL, 1.85 mmol) was added to a THF/Et_2_O (1:1, 0.4 mL) solution of **4** (92.3 mg, 109 µmol). After stirring for 12 h at room temperature, the solvent and residual cyclohexene were removed under reduced pressure, before the residue was dissolved in cyclohexane. After filtration, the solvent of the filtrate was evaporated under reduced pressure. The obtained orange solid was subjected to GPC (toluene) to give a mixture of **9a** and **9b** (66:34 ratio as judged by the ^1^H-NMR spectrum, 80.5 mg, 94.0 µmol, total yield: 86%). Continuous recrystallization of the mixture from hexane afforded a few crystals of **9a**, which could be isolated by the filtration. After evaporating the solvent of the filtrate, the residue was heated at 120 °C for a few hours under the reduced pressure. The following recrystallization of the resulting solids from pentane afforded a few crystals of **9b** in pure form. **9a**: orange solid. Mp. 174.1 °C (decomp). ^1^H-NMR (400 MHz, benzene-*d*_6_); δ 7.93 (d, *J* = 1.9 Hz, 4H), 7.53 (t, *J* = 1.9 Hz, 2H), 4.75 (s, 2H), 4.24 (br s, 2H), 4.11 (t, *J* = 1.9 Hz, 2H), 4.03 (s, 5H), 3.86 (br s, 5H), 1.99 (br s, 2H), 1.73 (br s, 2H), 1.49 (s, 36H), 1.26-1.27 (br, 6H); ^13^C-NMR (101 MHz, benzene-*d*_6_) δ 150.16, 139.86, 125.32, 120.40, 96.84, 75.04, 73.10, 72.66, 70.79, 69.16, 68.96, 66.73, 35.21, 31.91, 24.95, 22.66, 11.96; ^29^Si-NMR (79 MHz, benzene-*d*_6_) δ—67.6. HRMS (DART), *m/z*: Found: 856.38901 ([M^+^]), Calcd. for C_54_H_68_Fe_2_Si ([M^+^]): 856.38826. **9b**: orange solid. Mp. 151.2 °C (decomp). δ 7.94 (d, *J* = 1.8 Hz, 4H), 7.52 (t, *J* = 1.8 Hz, 2H), 4.73 (s, 2H), 4.16 (s, 5H), 4.10 (t, *J* = 1.8 Hz, 2H), 3.99 (t, *J* = 1.8 Hz, 2H), 3.82 (s, 5H), 2.06 (br s, 2H), 1.73–1.79 (br m, 4H), 1.52 (br s, 4H), 1.43 (s, 36H); ^13^C-NMR (101 MHz, benzene-*d*_6_) δ 150.25, 139.50, 125.27, 121.23, 97.72, 75.99, 73.13, 71.95, 71.12, 69.16, 68.23, 64.17, 35.13, 31.93, 23.92, 21.63, 14.47; ^29^Si-NMR (79 MHz, benzene-*d*_6_) δ—69.8. Continuous recrystallizations of a mixture of **9a** and **9b** from hexanes afforded a very small amount of single crystals of **9a** in pure form. Spectroscopic and XRD analyses of the single crystals thus obtained enabled us to identify **9a**. Silirane **9b** was identified based on the spectral data including ^29^Si NMR data, which was similar to those of **9a**.

#### 3.2.4. Thermolysis of Siliranes **9a** and **9b** in Cyclohexene

Cyclohexene (0.7 mL) was added to a mixture of **9a** and **9b** (94:6 ratio as judged by the ^1^H NMR spectrum). After heating at 75 °C for 30 min, the residual cyclohexene was removed under reduced pressure. The ^1^H NMR spectrum of the residue in C_6_D_6_ showed the signals for **9a** and **9b** (45:55 ratio).

#### 3.2.5. Thermolysis of Siliranes **9a** and **9b** in the Solid State

A mixture of the orange solids of **9a** and **9b** (74:26 ratio as judged by the ^1^H-NMR spectrum) was heated at 120 °C for 30 min under evacuation in an NMR tube (5 mm diameter) equipped with a J-Young© tap. The ^1^H-NMR spectrum of the residue in C_6_D_6_ showed the signals for **9a** and **9b** (45:55 ratio).

#### 3.2.6. Reaction of Siliranes **9a** and **9b** with Methanol

MeOH (0.40 mL) was added to a mixture of siliranes **9a** and **9b** (66:34, 24.0 mg, 28.0 µmol) at room temperature. After stirring for 72 h at room temperature, the solvent was removed under reduced pressure. The obtained residue was purified by column chromatography on silica gel (hexane/benzene = 4:1) to give **11** (6.3 mg, 7.8 µmol, 28%), **12** (2.3 mg, 2.8 µmol, 10%) and **13** (3.8 mg, 4.3 µmol, 15%) **11**: orange solid. Mp. 161–164 °C. ^1^H-NMR (400 MHz, benzene-*d*_6_) δ 7.98 (d, *J =* 1.8 Hz, 2H), 7.82 (d, *J =* 1.8 Hz, 2H), 7.51 (q, *J =* 1.8 Hz, 2H), 5.59 (s, 1H), 4.80 (d, *J =* 2.3 Hz, 1H), 4.74 (d, *J =* 2.3 Hz, 1H), 4.28 (s, 5H), 4.00 (td, *J =* 2.3, 1.1 Hz, 1H), 3.95 (td, *J =* 2.3, 1.1 Hz, 1H), 3.90–3.91 (m, 1H), 3.85 (s, 5H), 3.62 (s, 3H), 3.23 (t, *J =* 1.1 Hz, 1H), 1.44 (d, J = 1.3 Hz, 36H). HRMS (DART), *m/z*: Found: 806.32643 ([M^+^]), Calcd. for C_49_H_62_Fe_2_OSi ([M^+^]): 806.32688. **12**: orange solid. Mp. 67–71 °C. ^1^H-NMR (400 MHz, benzene-*d*_6_) δ 7.84 (d, *J =* 1.8 Hz, 4H), 7.51 (t, *J =* 1.8 Hz, 2H), 4.59 (s, 2H), 4.34 (s, 5H), 4.03 (t, *J =* 1.8 Hz, 2H), 4.01 (s, 5H), 3.82 (t, *J =* 1.8 Hz, 2H), 3.38 (s, 6H), 1.46 (s, 36H). HRMS (DART), *m/z*: Found: 836.33713 ([M^+^]), Calcd. for C_50_H_64_Fe_2_O_2_Si ([M^+^]): 836.33744. **13**: orange solid. Mp. 98–103 °C. ^1^H-NMR (400 MHz, benzene-*d*_6_) δ 7.73 (d, *J =* 1.8 Hz, 2H), 7.65 (d, *J =* 1.8 Hz, 2H), 7.47–7.48 (m, 2H), 4.54 (d, *J =* 2.3 Hz, 1H), 4.52 (d, *J =* 2.3 Hz, 1H), 4.27 (s, 5H), 4.08–4.10 (m, 3H), 4.01 (s, 5H), 3.94 (s, 1H), 3.29 (s, 3H), 2.27 (br d, *J =* 13.8 Hz, 1H), 2.16 (br d, *J =* 13.8 Hz, 1H), 1.82 (br d, *J =* 13.8 Hz, 1H), 1.75 (br d, *J =* 13.8 Hz, 2H), 1.55–1.60 (br, 1H), 1.45 (d, *J =* 1.8 Hz, 36H), 1.40–1.41 (br, 1H), 1.39–1.39 (br, 1H), 1.08–1.30 (br, 3H). HRMS (DART), *m/z*: Found: 888.40586 ([M^+^]), Calcd. for C_55_H_72_Fe_2_OSi ([M^+^]): 888.40513.

#### 3.2.7. Reaction of Siliranes **9a** and **9b** with 2,3-Dimethyl-1,3-butadiene

2,3-Dimethyl-1,3-butadiene (0.50 mL, 4.40 mmol) was added to a mixture of **9a** and **9b** (25:86 ratio, 5.7 mg, 6.7 µmol). After stirring for 41 h at 60 °C, the residual 2,3-dimethyl-1,3-butadiene was removed under reduced pressure to give **14** (quant. as judged by the ^1^H-NMR spectrum). The obtained orange solid of **14** was purified by column chromatography on silica gel (hexane/benzene = 4:1) to give **14** in pure form (2.5 mg, 2.9 µmol, 44%). **14**: orange solid. Mp. 242 °C (decomp). ^1^H-NMR (400 MHz, benzene-*d*_6_) δ 7.70 (d, *J =* 1.8 Hz, 4H), 7.50 (t, *J =* 1.8 Hz, 2H), 4.53 (s, 2H), 4.36 (s, 5H), 4.17 (t, *J =* 1.8 Hz, 2H), 4.06 (t, *J =* 1.8 Hz, 2H), 3.86 (s, 5H), 2.02 (d, *J =* 17.6 Hz, 2H), 1.77 (m, 8H), 1.40 (s, 36H) ^29^Si-NMR (79 MHz, benzene-*d*_6_) δ 2.15. HRMS (DART), *m/z*: Found: 856.38994 ([M^+^]), Calcd. for C_54_H_68_Fe_2_Si: 856.38826.

#### 3.2.8. Characterization of Fc*(Fc)Si(OH)_2_

Orange solid. Mp. 144-151 °C. ^1^H-NMR (400 MHz, benzene-*d*_6_) δ 7.84 (d, *J* = 1.9 Hz, 4H), 7.50 (t, *J* = 1.9 Hz, 2H), 4.57 (s, 2H), 4.34 (s, 5H), 3.99 (q, *J* = 1.9 Hz, 7H), 3.85 (t, *J* = 1.9 Hz, 2H), 2.68 (s, 2H), 1.40 (s, 36H). HRMS (APCI), *m/z*: Found: 808.31153 ([M^+^]), Calcd. for C_48_H_60_Fe_2_O_2_Si ([M^+^]): 808.30614. 

### 3.3. Computational Methods

The level of theory and the basis sets used for the structural optimization are given in the main text. Frequency calculations confirmed minimum energies for all optimized structures. All calculations were carried out on the *Gaussian 16* (Revision C.01) program package [53]. Computational time was generously provided by the Supercomputer Laboratory in the Institute for Chemical Research of Kyoto University.

### 3.4. X-ray Crystallographic Analysis

Single crystals of **5**, **9a**, **10**, **11**, **13**, **14**, and Fc*FcSi(OH)_2_ were obtained by recrystallization from Et_2_O (**5**, **9a**,**10**, **14**, Fc*FcSi(OH)_2_) at room temperature or from Et_2_O/MeOH (**11**, **13**) at −30 °C. Intensity data for **9a** were collected on a RIGAKU Saturn70 CCD(system) with VariMax Mo Optics using Mo-Kα radiation (λ = 0.71073 Å), while those for others were collected at the BL02B1 beamline of SPring-8 (2018A1167, 2018B1668, 2018B1179, 2019A1057, 2019A1677, 2019B1129, 2019B1784, 2020A1056, 2020A1644, 2020A1650, 2020A0834) on a PILATUS3 X CdTe 1M camera using synchrotron radiation (λ = 0.4148 Å). The structures were solved using SHELXT-2014 and refined by a full-matrix least-squares method on *F*^2^ for all reflections using the programs of SHELXL-2016 [54,55]. All non-hydrogen atoms were refined anisotropically, and the positions of all hydrogen atoms were calculated geometrically and refined as riding models. Supplementary crystallographic data were deposited at the Cambridge Crystallographic Data Centre (CCDC) under deposition numbers CCDC-2044338-2044344 and can be obtained free of charge from via www.ccdc.cam.ac.uk/data_request.cif.

Crystallographic data for **5** (CCDC-2044338): C_58_H_66_Fe_2_Si, FW 902.89, crystal size 0.01 × 0.01 × 0.01 mm^3^, temperature −180 °C, *λ* = 0.4148 Å, triclinic, space group *P*–1 (#2), *a* = 12.072(5) Å, *b* = 12.297(5) Å, *c* = 16.770(7) Å, *α* = 84.253(5)°, *β* = 80.861(5)°, *γ* = 89.400(5)°, *V* = 2445.4(18) Å^3^, *Z* = 2, *D*_calcd_ = 1.226 g cm^−3^, *μ* = 0.158 mm^−1^, *θ*_max_ = 15.729°, 57,864 reflections measured, 11360 independent reflections (*R*_int_ = 0.0717), 692 parameters refined, GOF = 1.029, completeness = 99.4%, *R*_1_ [*I* > 2*σ*(*I*)] = 0.0651, w*R*_2_ (all data) = 0.1685, largest diff. peak and hole 1.262 and –1.498 e Å^–3^. **9a** (CCDC-2044339): C_54_H_68_Fe_2_Si, FW 856.87, crystal size 0.15 × 0.10 × 0.08 mm^3^, temperature –170 °C, *λ* = 0.71073 Å, triclinic, space group *P*–1 (#2), *a* = 10.1310(3) Å, *b* = 15.7957(4) Å, *c* = 16.3047(4) Å, *α* = 67.199(2)°, *β* = 76.969(2)°, *γ* = 76.374(2)°, *V* = 2311.22(11) Å^3^, *Z* = 2, *D*_calcd_ = 1.231 g cm^–3^, *μ* = 0.688 mm^–1^, *θ*_max_ = 30.835°, 41,513 reflections measured, 9056 independent reflections (*R*_int_ = 0.0429), 551 parameters refined, GOF = 1.045, completeness = 99.6%, *R*_1_ [*I* > 2*σ*(*I*)] = 0.0540, w*R*_2_ (all data) = 0.1443, largest diff. peak and hole 1.694 and –1.089 e Å^–3^. **10** (CCDC-2044340): C_48_H_60_Fe_2_OSi, FW 792.75, crystal size 0.02 × 0.01 × 0.01 mm^3^, temperature –180 °C, *λ* = 0.4148 Å, triclinic, space group *P*–1 (#2), *a* = 11.551(5) Å, *b* = 13.356(6) Å, *c* = 15.732(7) Å, *α* = 104.761(6)°, *β* = 108.360(6)°, *γ* = 100.704(6)°, *V* = 2131.9(17) Å^3^, *Z* = 2, *D*_calcd_ = 1.235 g cm^–3^, *μ* = 0.177 mm^–1^, *θ*_max_ = 15.694°, 50,376 reflections measured, 9876 independent reflections (*R*_int_ = 0.0869), 491 parameters refined, GOF = 1.016, completeness = 99.6%, *R*_1_ [*I* > 2*σ*(*I*)] = 0.0449, w*R*_2_ (all data) = 0.1211, largest diff. peak and hole 0.549 and –0.861 e Å^–3^. **11** (CCDC-2044341): C_49_H_62_Fe_2_OSi, FW 806.78, crystal size 0.01 × 0.01 × 0.01 mm^3^, temperature –180 °C, *λ* = 0.4148 Å, triclinic, space group *P*–1 (#2), *a* = 9.748(8) Å, *b* = 12.421(10) Å, *c* = 18.807(15) Å, *α* = 78.213(8)°, *β* = 78.790(11)°, *γ* = 85.260(9)°, *V* = 2185(3) Å^3^, *Z* = 2, *D*_calcd_ = 1.227 g cm^–3^, *μ* = 0.173 mm^–1^, *θ*_max_ = 15.765°, 52,665 reflections measured, 9471 independent reflections (*R*_int_ = 0.0912), 553 parameters refined, GOF = 1.040, completeness = 92.1%, *R*_1_ [*I* > 2*σ*(*I*)] = 0.0879, w*R*_2_ (all data) = 0.2863, largest diff. peak and hole 1.659 and –1.131 e Å^–3^. **13** (CCDC-2044342): C_59_H_82_Fe_2_O_2_Si, FW 963.04, crystal size 0.01 × 0.01 × 0.01 mm^3^, temperature –180 °C, *λ* = 0.4148 Å, triclinic, space group *P*–1 (#2), *a* = 10.29(5) Å, *b* = 15.09(7) Å, *c* = 16.60(8) Å, *α* = 104.05(4)°, *β* = 96.46(5)°, *γ* = 92.32(5)°, *V* = 2478(21) Å^3^, *Z* = 2, *D*_calcd_ = 1.291 g cm^–3^, *μ* = 0.158 mm^–1^, *θ*_max_ = 9.863°, 14,683 reflections measured, 2889 independent reflections (*R*_int_ = 0.2584), 577 parameters refined, GOF = 1.049, completeness = 98.3%, *R*_1_ [*I* > 2*σ*(*I*)] = 0.1088, w*R*_2_ (all data) = 0.3010, largest diff. peak and hole 0.320 and –0.426 e Å^–3^. Only preliminary data have been obtained, since the single crystals of this compound have been obtained only in very poor quality despite several attempts of careful recrystallization probably due to the partial hydrolysis of the Si-OMe moiety. **14** (CCDC-2044343): C_54_H_68_Fe_2_Si, FW 856.87, crystal size 0.01 × 0.01 × 0.01 mm^3^, temperature –180 °C, *λ* = 0.4148 Å, triclinic, space group *P*–1 (#2), *a* = 10.546(10) Å, *b* = 12.346(12) Å, *c* = 18.329(19) Å, *α* = 92.52(2)°, *β* = 97.949(12)°, *γ* = 99.949(16)°, *V* = 2322(4) Å^3^, *Z* = 2, *D*_calcd_ = 1.225 g cm^–3^, *μ* = 0.164 mm^–1^, *θ*_max_ = 13.916°, 37,917 reflections measured, 7520 independent reflections (*R*_int_ = 0.0674), 529 parameters refined, GOF = 1.031, completeness = 99.3%, *R*_1_ [*I* > 2*σ*(*I*)] = 0.0437, w*R*_2_ (all data) = 0.1260, largest diff. peak and hole 0.771 and –0.648 e Å^–3^. Fc*FcSi(OH)_2_ (CCDC-2044344): C_48_H_60_Fe_2_O_2_Si, FW 808.75, crystal size 0.25 × 0.20 × 0.10 mm^3^, temperature –170 °C, *λ* = 0.71075 Å, triclinic, space group *P*–1 (#2), *a* = 11.4879(2) Å, *b* = 13.2748(1) Å, *c* = 15.6366(2) Å, *α* = 104.3460(1)°, *β* = 108.2530(1)°, *γ* = 101.1560(1)°, *V* = 2096.58(5) Å^3^, *Z* = 2, *D*_calcd_ = 1.281 g cm^–3^, *μ* = 0.758 mm^–1^, *θ*_max_ = 26.999°, 39,810 reflections measured, 9082 independent reflections (*R*_int_ = 0.0365), 718 parameters refined, GOF = 1.127, completeness = 99.2%, *R*_1_ [*I* > 2*σ*(*I*)] = 0.0365, w*R*_2_ (all data) = 0.0826, largest diff. peak and hole 0.395 and –0.437 e Å^–3^.

## 4. Conclusions

Bis(ferrocenyl)siliranes **9a** and **9b** were prepared by the reduction of the corresponding dichlorosilane with lithium naphthalenide in the presence of an excess of cyclohexene. Siliranes **9a** and **9b** are appropriate precursors for bis(ferrocenyl)silylenes upon heating under mild conditions, i.e., they can be considered as bottleable synthetic precursors for silylenes. Further investigations into the creation of redox-active organosilicon compounds that bear ferrocenyl moieties by using siliranes **9a** and **b** as silylene precursors are currently in progress in our laboratory.

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
