# Peer review of "Generation of Bis(ferrocenyl)silylenes from Siliranesâ€"

_molecules, 2020, doi:10.3390/molecules25245917_

Round 1
Reviewer 1 Report
This manuscript by Sasamori et al. describes the formation of intermediate silylenes by thermolysis of a silyrene derivative. The formation of the silylene is confirmed by trapping experiments and their products are adequately characterized. In my opinion, the manuscript can be recommended for publication with only minor changes.
Compounds I-V in Scheme 1 are not explicitly mentioned in the Introduction.
References to programs used to solve and refine the crystal structures must be added.
Experimental HRMS data for 13 differ unacceptably much from the expected value.
Typing errors:
Line 15: redox-active.
Line 19: bottleable.
Line 219: dispersion (in what?).
Line 245: NMR tube (5 mm diameter).
Line 137: although they are (instead of albeit that they are).
Line 435: poor; line 436: despite (instead of even though).
Reference 13, crystallographic data: unusual symbol after mu (absorption coefficient).
Entire manuscript: benzene-d6 (not BENZENE-D6).
Author Response
Please see the attached pdf file.

Reviewer 2 Report
The manuscript by Pan et al. introduces new ferrocenyl-substituted silylene precursors and reactivity thereof. It is an interesting addition to the chemistry of low-valent reactive silicon species, which could be seen as potential building blocks for redox-active organosilicon compounds. From this point of view, it fits the journal scope. The work presented has been obviously conducted in a well-organized and rigorous manner and is presented clearly. The manuscript is also well written with a minimum of typos. I have only several rather technical or formal recommendations as listed below. In conclusion, I recommend this paper to be published after minor revision and congratulate the authors for their nice work.
Corrections and comments:
abstract – line 15: “redoxactive” – “redox-active”
line 19: “bottleable”
page 2, chart 1: There could be literature references to the individual examples of known silylenes I-V
page 3, line 112: ratio in favor of 9b (9a : 9a…should be “9a:9b”
page 4, line 133: “treatment of methanol solution…with methanol” – reformulate so that it would be clear that the mixture of isomers was actually dissolved in excess of methanol
page 5, scheme 4: there could be an information about stoichiometry of the reaction with methanol (it was in excess – as discussed above)
line 162: “naphtalide” – “naphthalenide”
Author Response
Please see the attached pdf file.

Reviewer 3 Report
This paper describes the design and synthesis of a bis(ferrocenyl)silirane as a precursor for a bis(ferrocenyl)silylene, which could be used as a building block for redox-active organosilicon compounds. The authors suggest that these organosilicon compounds can be bottleable precursor for the bis(ferrocenyl)silylene under mild conditions. The manuscript is clear and well-written. The authors describe the bis(ferrocenyl)dichlorosilane (4) as a stable compound and claim a full spectroscopic characterization, however the 29Si-NMR characterization was not included. The thermolysis of 9a/9b in presence of 2,3-butadiene gives evidences of the transient formation of the silylene intermediate. It would have been very interesting to know the redox behavior of these ferrocenyl-silicon compounds and correlate if the presence of the ferrocenyl moiety modifies (or not) their stability and reactivity.
Some minor correction:
Chart 1: it must be isolable
Page 3, line 112: verify the identity of (9a : 9a)
Author Response
Please see the attached pdf file.

Reviewer 4 Report
This manuscipt by the Sasamori group is a classical contribution in organoelement chemistry towards a FcFc'-silylene precursor as a potential synthon for redox-responsive oligo and polysilanes. Overall, the work is well performed with good literature coverage and sound conclusions. The 2+1 silirane cycloadducts of the in situ generated silylene with cyclohexene are interesting synthons for releasing the reactive FcFc'silylene via cycloreversion at mild conditions. I recommend publication of the paper as it is without any changes.
Author Response
We greatly appreciate the very positive comments. We are very pleased and encouraged to see them.